# Male and Female Mitochondria Respond Differently after Exercising in Acute Hypoxia

**DOI:** 10.3390/biomedicines11123149

**Published:** 2023-11-26

**Authors:** Ylenia Lai, Francesco Loy, Michela Isola, Roberta Noli, Andrea Rinaldi, Carla Lobina, Romina Vargiu, Flaminia Cesare Marincola, Raffaella Isola

**Affiliations:** 1Department of Biological Sciences, University of Cagliari, Cittadella Universitaria di Monserrato, 09042 Monserrato, Italy; 2Neuroscience Institute, Division of Cagliari, National Research Council of Italy, Cittadella Universitaria di Monserrato, 09042 Monserrato, Italy; 3Department of Chemical and Geological Sciences, University of Cagliari, Cittadella Universitaria di Monserrato, 09042 Monserrato, Italy

**Keywords:** hypoxia, mitochondrial bioenergetics, heart, brain

## Abstract

The use of hypoxic devices among athletes who train in normobaric hypoxia has become increasingly popular; however, the acute effects on heart and brain metabolism are not yet fully understood. This study aimed to investigate the mitochondrial bioenergetics in trained male and female Wistar rats after acute hypoxia training. The experimental plan included exercising for 30 min on a treadmill in a Plexiglas cage connected to a hypoxic generator set at 12.5% O_2_ or in normoxia. After the exercise, the rats were sacrificed, and their mitochondria were isolated from their brains and hearts. The bioenergetics for each complex of the electron transport chain was tested using a Clark-type electrode. The results showed that following hypoxia training, females experienced impaired oxidative phosphorylation through complex II in heart subsarcolemmal mitochondria, while males had an altered ADP/O in heart interfibrillar mitochondria, without any change in oxidative capacity. No differences from controls were evident in the brain, but an increased electron transport system efficiency was observed with complex I and IV substrates in males. Therefore, the study’s findings suggest that hypoxia training affects the heart mitochondria of females more than males. This raises a cautionary flag for female athletes who use hypoxic devices.

## 1. Introduction

When people enter high altitudes where oxygen concentration decreases, their ability to perform exercise declines [1,2]. This is due to the low partial pressure of oxygen (PO_2_), which reduces maximal aerobic power, and, in response, the human body increases pulmonary ventilation. However, chronic exposure (7 to 28 days) to low levels of air oxygenation results in an improvement in physical performance. This is because chronic hypoxia induces several physiological adaptations, including augmented hemoglobin and polycythemia, greater pulmonary ventilation that leads to increased maximal aerobic power, and lower neuromuscular fatigability [2]. For this reason, chronic hypoxia has been extensively studied for unraveling its multifaceted physiological impacts on human performance [1,2,3,4].

To enhance their performance, athletes often use hypoxic gas generators for training. However, they exercise in normobaric hypoxia without proper acclimatization or knowledge of the potential adverse effects on their cardiovascular system [5,6]. Moreover, studies on the effect of low oxygen on the body have been primarily limited to healthy or athletic subjects [2,7]. Given that studies involving humans are limited in their ability to collect comprehensive physiological parameters, the supplementation of such information with translational studies becomes imperative. The effects of exposing experimental animal models to low oxygen levels have been broadly studied, encompassing both short- (6 to 48 h) and long-term (7 to 28 days) durations. In the case of short-term exposure, the focus has mainly been on animals at rest rather than during exercise [8,9,10]. It is worth mentioning that a correct acute (0.5–1 h) hypoxic experimentation is lacking. Past studies on the effect of hypoxia on heart or skeletal muscle mitochondria have focused on long-term exposure to hypoxia [3,4,11,12]. On the other hand, studies on the effect of hypoxia on brain mitochondria were tested as short-term (3 or 48 h) or after ischemia-reperfusion injury, with conflicting results [10,13,14]. However, no previous research has delved into the acute effects of hypoxia on heart and brain mitochondria in both sexes following exercise.

Our investigation targeted male and female trained rats exposed to 30 min of hypoxia during treadmill exercises, compared with sessions in normoxic conditions. The level of the chosen hypoxia corresponded to an altitude of 4000 m (i.e., an oxygen amount of 12.5% in the inhaled air). Our research aim was to investigate the reactive oxygen species (ROS) production and the expression of G0S2, a mitochondrial factor induced by hypoxia, which could affect mitochondria activity. The results revealed distinct behaviors of male and female mitochondria in response to hypoxia during exercise.

## 2. Materials and Methods

This study was approved by the Committee on Animal Use and Care of University of Cagliari (CeSASt) in accordance with the Italian Guidelines for the use of laboratory animals, which conform with the guidelines for care and use of experimental animals issued by the Italian Ministry of Health (D.L. 116/92) and by the EU Directive (2010/63/EU) and the Guide for Care and Use of Laboratory Animals, adopted by the NIH, USA (8th edition, 2011). Furthermore, the experimental plan was approved by the Italian Ministry of Health (number of approval n. 579/2020-PR).

### 2.1. Animals and Training Design

Twenty-eight male and twenty-five female Wistar Han rats (Envigo, Udine, Italy) were used. They were housed at 3–4 animals per cage, held in a 12 h light cycle (6 am light and 6 pm dark), and they received water and food ad libitum. Rats were trained 5 days a week for 5 weeks (aged from 7 to 11 weeks old for males and from 9 to 13 weeks old for females), in order to mimic trained healthy individuals. The training program consisted in running gradually on a treadmill (2Biological Instruments LE8710, Varese, Italy). In the first two weeks, the rats underwent training sessions involving running speeds ranging from 16 to 35 cm/s, durations spanning 25 to 60 min, and inclines varying from no incline to planes with a 5° slope. In the 3rd to 5th weeks, the final training protocol was applied: one hour of total running time, with a 5° slope, 38 min of which at full speed (35 cm/s), 17 min at a medium pace (25 cm/s), 5 min at a slow pace (17 cm/s), and then 1 min to stop. At the end of the training, a small piece (250 mg) of white chocolate was given as a reward, since it was reported to better motivate rats for running activity [15].

### 2.2. Experimental Protocol

On the day of the experiment, rats were allowed to run on the treadmill with 5° slope for initial 5 min in normoxia at 15 cm/s, followed by 30 min at 35 cm/s in hypoxia (12.5% oxygen concentration, the same as at 4000 m). Hypoxia was obtained by connecting a Plexiglas box, containing the treadmill, to a hypoxic gas generator (Everest Summit II Generator, Hypoxico, New York, NY, USA). Preliminary experiments verified that inside the Plexiglas chamber, the desired oxygen concentration was reached within a few minutes. The experiment was carried out under normobaric conditions. As soon as the time was finished, rats were euthanized via decapitation and the brain and heart were quickly removed and placed in an appropriate ice-cold buffer.

### 2.3. Mitochondria Isolation and Bioenergetics

Mitochondria brain isolation was performed with some modifications based on the procedures outlined in [16]. Following extraction, brains were placed in ice-cold MSM (220 mM Mannitol, 70 mM Sucrose, and 5.0 mM MOPS, pH 7.4). The cerebellum was discarded and the anterior half of the brain, including the forebrain and parietal cortex, as well as part of the corpus callosum and the basal ganglia, was furtherly processed. The tissue was then minced with small scissors, incubated for 30 s with 5 mg/g of tissue Subtilisin in MSM + 2 mM EDTA (1 mL/g of tissue), homogenized with a Teflon homogenizer, and then MSM + 2 mg/mL BSA was added to stop the reaction. Samples were then centrifuged at 12,000× *g* for 10 min, and pellets were resuspended in MSM + BSA. Subsequently, they were centrifuged again at 350× *g* for 10 min. The supernatants from this step were centrifuged at 7000× *g* for 10 min, and pellets were resuspended twice in 4 mL MSM + BSA followed by centrifugation at 7000× *g* for 10 min. The last pellet was then washed twice in MSM each time in half the volume and centrifuged at 7000× *g* for 10 min. The final pellet was resuspended in about 0.2 mL MSM.

In the heart, we isolated subsarcolemmal (SSM) and interfibrillar (IFM) mitochondria following the method of Palmer and colleagues with modifications introduced by Rosca et al. [17,18]. Briefly, hearts were placed in ice-cold Chappell–Perry buffer (CP1) (100 mM KCl, 50 mM MOPS, 5 mM MgSO_4_, 1 mM ATP, and 1 mM EGTA), then atria were discarded and the tissue was minced and incubated for 10 min with CP1 without EGTA containing 25 mg/g of heart collagenase; CP2 was then added (buffer CP1 plus 0.2% defatted BSA) to stop the reaction; and then the pellet was centrifuged for 10 min at 580× *g*, resuspended in CP2, and homogenized with a Potter Elvehjem homogenizer to free subsarcolemmal mitochondria. The homogenate was then centrifuged at 580× *g*, the supernatant was put aside, the pellet was washed with CP2 and centrifuged again at 580× *g*, the pellet was then put aside, and the resulting supernatant was combined with the first and centrifuged at 3100× *g* to obtain the first pellet of SSM mitochondria that was later washed several times via centrifugation and then suspended in KME (100 mM KCl, 50 mM MOPS, and 0.5 mM EGTA, pH 7.46) for storage. The remaining pellet from the last 580 g centrifugation was then treated with 5 mg/g of heart trypsin in CPT1 without EGTA, resuspended using a Potter–Elvehjem homogenizer, and then incubated for 8 min before CPT2 plus a trypsin inhibitor was added (to stop the reaction), after which it was homogenized again and centrifuged at 580× *g* for 10 min. The resulting supernatant was centrifuged at 3100× *g* and the following pellet contained IFM mitochondria and was washed several times and lastly suspended in KME. In SSM and IFM samples, mitochondrial protein concentration was determined using the BCA (Bicinchoninic acid) assay with bovine serum albumin as standard.

Mitochondrial bioenergetics was assessed at 37 °C using a Clark-type electrode (Oxytherm, Hansatech Instruments Ltd., Norfolk, UK) in the respiration buffer (100 mM KCl, 50 mM Mops, 1 mM EGTA, 5 mM KH_2_PO_4_, and 1 mg/mL defatted BSA, pH 7.4). The following substrates were added to test oxidative phosphorylation, with the following final concentrations [19]: 20 mM glutamate/10 mM malate (substrates of complex I), 7 µM Rotenone/20 mM succinate (substrates of complex II), 7 µM Rotenone/1 mM durohydroquinone (complex III), 7 µM Rotenone/500 µM tetramethyl-p-phenylenediamine dihydrochloride (TMPD)/5 mM ascorbate, and 2.5 mM ascorbate alone (substrate for complex IV) or 20 µM Palmitoyl Coenzyme A/5 mM malate/5 mM carnitine (lipid substrates). Subsequently, to assess state 3 (active state) 200 µM ADP or to test state 4 (resting state) 0.5 mg/mL oligomycin were added to the chamber. To measure maximal oxidative capacity, 2 mM ADP was added to mitochondria after each substrate, followed by 200 µM dinitrophenol (DNP; for the uncoupled respiration or maximal activity of electron transport system). The amount of mitochondria tested each time was 0.25 mg/500 µL with glutamate/malate and Palmitoyl CoA and 0.125 mg/500 µL with succinate, durohydroquinone, and TMPD. The number of samples was 13–15 for males and 12–13 for females. There were some exceptions. As substrate for complex II, in males, succinic acid (with very poor solubility) was initially used instead of sodium succinate. The bioenrgetic assessments were more reliable with the latter, so the number for rotenone/succinate samples is lower in males. For brains, *n* values were 10 in controls and 17 in hypoxic samples. This is because we utilized some of the brains from parallel physiology experiments that were performed on different rats. Moreover, sometimes, when statistics were applied, outliers were canceled and the *n* varied accordingly. Outliers could arise from incorrect pipetting, altered substrates, or the incorrect functioning of the oxygraph.

### 2.4. Immunoblotting

Isolated SSM and IFM mitochondria from male or female hypoxic and control rats were diluted 1:3 in loading buffer (4% SDS, 20% glycerol, 160 mM dithioerythrol, 125 mM Tris-Cl, bromophenol blue 0.004%, pH 6.8) [20,21]. Samples were then sonicated for 3 min and subsequently heated for 10 min at about 76–78 °C for denaturation. The samples’ proteins were then separated via electrophoresis on 4–20% Mini-PROTEAN^®^ TGX™ precast polyacrylamide gels (Biorad, Hercules, CA, USA), and then transferred on PVDF membranes, which were blocked for 2 h with 5% milk in Tris-buffered saline with 0.1% Tween 20 (TBS-T). Incubation with primary antibodies followed overnight at 4 °C. The following primary antibodies were used: rabbit G0S2 antibody (G0/G1 switch gene 2, 1:1000, Proteintech, Leon-Rot, Germany) and goat anti-4-hydroxy-2-nonenal antibody (HNE, 1:5000, Merck-Millipore, Darmstadt, Germany). Secondary antibodies (goat anti-rabbit peroxidase conjugate, 1:2000, and rabbit anti-goat peroxidase conjugate, 1:500, both by Merck-Millipore, Darmstadt, Germany) were incubated for 1 h at room temperature. The detection of proteins was performed using the ECL Prime Chemiluminescence kit (GE HealthCare, Little Chalfont, UK) and images were acquired with a Fujifilm Luminescent Image Analyzer LAS4000 System (Fujifilm, Tokyo, Japan). Immunoreactive bands were analyzed for densitometry with Image Studio Lite Software Ver. 5 (LI-COR, Lincoln, NE, USA). Proteins quantification was expressed as the relative intensity of protein signals normalized to the expression of the mitochondrial housekeeping gene COX IV (rabbit anti-COX-IV, Cytochrome c Oxidase subunit IV, 1:9000, Life Technologies, ThermoFisher, Waltham, MA, USA). Using Image Studio Light, we performed a contextual background subtraction of each measurement by calculating the median value in a border of 3 pixels around the selected shape, which was then multiplied by the area of the shape, and the resulting value was subtracted by the total of the intensity values obtaining the correct signal values.

### 2.5. Statistics

Statistics was obtained using Sigma Plot software 11.0 (Systat software Inc., San Jose, CA, USA). The normality of data was verified in order to choose the appropriate test between a *t*-test (for parametric data) or the Mann–Whitney U test (for non-parametric data). The statistical level was fixed at *p* < 0.05.

## 3. Results

### 3.1. Training

The animals (aged 7–13 weeks during the training period) demonstrated good tolerance to the five-week training regimen. Generally, males exhibited a tendency to be less active than females, while the latter group displayed a greater fondness for consuming chocolate compared to the opposite sex. Despite the exercise routine, the weights of both males and females remained within their expected range for their age (Figure 1, and for a comparison check, https://www.inotivco.com/model/rcchan-wist, accessed on 30 August 2023), indicating that moderate exercise did not influence their body weights.

### 3.2. Mitochondria Bioenergetics

The yields from mitochondrial isolation, measured as milligrams of total mitochondrial proteins per gram of original tissue, showed no statistically significant differences between hypoxic and control animals, for both heart and brain mitochondria (Table 1). We, therefore, observed no indication that mitophagy happened during hypoxia.

#### 3.2.1. Heart Mitochondria

Oxidative phosphorylation (OXPHOS) was tested with substrates for complex I, II, III, and IV and with a lipid substrate in heart controls or hypoxic SSM or IFM mitochondria for both males and females. Most of the tested substrates did not display any differences in comparison to control groups, whether in IFM or SSM OXPHOS (Appendix A). The sole observed mitochondrial impairment was noted in females’ SSM at the complex II-linked OXPHOS, where the rate of state 3 was reduced after training in hypoxia (Figure 2).

In male IFM samples, a reduction in the ADP/O ratio was observed when using complex I substrates. The ADP/O ratio is the ratio between the quantity of nanomoles of ADP introduced into the chamber and the number of nanomoles of oxygen consumed to produce ATP, specifically, the amount of oxygen required to phosphorylate an individual ADP molecule. This decrease suggests a rise in oxygen consumption per molecule of ATP produced, given that the amount of ADP remains constant. Conversely, in females’ IFM, the ADP/O ratios for both control and hypoxic samples displayed similarities (as depicted in Figure 3). Furthermore, in hypoxic male IFM, the substrate for complex I showed an increased respiratory control ratio (RCR, state 3/state 4). This parameter is usually associated with a good coupling of mitochondrial preparations, even if it can sometimes be associated with its dysfunction [20]. Given the absence of an observed increase in state 3 or a decrease in state 4, it might be that there was a slight increased oxygen consumption rate with the complex I substrate in hypoxic mitochondria (Figure 3). We hypothesized that this tendency to an increased oxygen consumption might have arisen from a more efficient F0F1 ATPase or from increased ROS production.

#### 3.2.2. Brain Mitochondria

In both male and female brain mitochondria, the OXPHOS remained unchanged across all tested substrates after exposure to hypoxia (Appendix A). The only difference observed in males was an increased rate of oxygen consumption with the complex II and complex IV substrates. This increase was seen after the addition of the uncoupler following the addition of 2 mM ADP. Specifically, within hypoxic male uncoupled mitochondria, oxygen consumption exceeded that of control groups (as shown in Figure 4). This result suggests a probable enhanced electron transport system (ETS) efficiency after hypoxia in males. Conversely, females did not show any differences with the same substrates and uncoupler (Figure 4).

### 3.3. Immunoblotting

To identify the presence of ROS or the enhanced expression of G0S2 (a factor known to increase ATP-synthase efficiency under hypoxia), we performed immunoblotting on proteins extracted by isolated IFM or SSM male or female mitochondria.

Protein expression for G0S2 increased both in females and males IFM after exercising in hypoxia, as compared to controls, whereas its production did not vary in SSM of either sexes (Figure 5).

When we investigated the HNE-conjugated proteins’ expression (resulting in lipid peroxidation, in turn due to ROS production), both female and male hypoxic SSM showed an increased expression of HNE-proteins compared to controls, which reached statistical significance in male SSM only. The IFM of both sexes did not show any differences.

## 4. Discussion

In the present study, we investigated the effects of acute hypoxia during exercise in trained male and female rats. To the best of our knowledge, this is the first study to examine the real acute normobaric hypoxia across both sexes. Most previous studies on acute hypoxia were carried out over a long period of time, sometimes up to 48 h [9,10,11], disregarding the physiological changes that occur in early hypoxia. Despite being underestimated for a long time, acute hypoxic effects on the heart and brain are important not only for basic physiologic knowledge but also as a warning to possible physical injuries after the use of hypoxic devices during training.

The most striking result of our work is that heart and brain mitochondrial impairment is different between male and female rats. In female hearts, complex II-linked OXPHOS is impaired in SSM only, whilst male heart mitochondria did not show any defective OXPHOS. On the other hand, within IFM, the only difference in the controls was noticed in males regarding the values of ADP/O and RCR related to the complex I substrate. Although this alteration may suggest a certain elevation in oxygen consumption during OXPHOS, the change did not effectively correlate to substantial modifications in the oxidative capacity of OXPHOS via complex I.

A further effect of hypoxia was enhanced ROS production, albeit solely in SSM, reaching statistical significance exclusively in males. On the other hand, the protein expression of G0S2, a protein factor expressed during hypoxia that may enhance ATPase activity, was increased both in female and male IFM, showing specificity for these mitochondria. These findings underscore the differing roles and sensitivities to hypoxia exhibited by these two subsets of heart mitochondria independently of sex. Moreover, the brain mitochondria of males displayed an augmented efficiency in the ETS when challenged through complex II and IV following acute hypoxia.

Studies on mitochondrial bioenergetics after acute hypoxia in the heart or brain are scarce. A study on mice hearts demonstrated that after 24 h of acute hypoxia (13% O_2_), there was a reduction in OXPHOS rate through complex I and with lipid substrates [9]. It was also shown that chronic hypoxia in male rats (3 weeks at 11% O_2_) affects SSM and IFM populations of heart mitochondria differently [22]. In that study, both SSM and IFM had lower respiration with lipid substrates and pyruvate (complex I), but only SSM displayed reduced respiration with substrates for complex II, III, and IV. In another recent report, where hypoxia-resistant and hypoxia-sensitive rats were exposed to one hour of hypoxia, the expression of the catalytic subunits of several enzymes of the electron transport chain was evaluated. While the authors concluded that in early hypoxia complex II activity is enhanced, arriving at this conclusion without conducting bioenergetics experiments presents challenges [23].

The majority of investigations regarding the effects of hypoxia have predominantly focused on long-term consequences and human skeletal muscle mitochondria following acclimatization to high altitudes. In these studies, it was found that, in humans, fatty acid oxidation and complex II-linked respiration decreased [4], or, among other things, the expression of subunits of complex II and III was reduced [24]. In skeletal muscle tissue, the decreased expression of complex I and complex IV and mitochondrial SSM number were observed after a long-term stay at extremely high altitudes [3]. It is important to note that the results mentioned earlier were obtained only from male subjects. In contrast, a study conducted on trained female rats in chronically hypoxic conditions found that respiration through complex I and with lipid substrates was reduced after hypoxia, despite the expected improvement from training [25].

We showed that the sole OXPHOS deficiency occurring in the heart following a 30 min hypoxic period is observed within the mitochondria of female SSM fractions, specifically affecting complex II. This result addresses a higher sensitivity of cardiac complex II to acute hypoxic conditions and highlights the higher sensitivity of female SSM in comparison to male SSM. As reported above, chronic hypoxia was the cause of decreased respiration at complex II [22], but in our experiments, the observed long-term hypoxia impairments at complexes III and IV were absent. On the other hand, in male SSM no effect was recorded in OXPHOS with different substrates after hypoxia. This result is rather remarkable, since in the literature, it seems that female cardiac mitochondria are protected from the adverse effects of cardiovascular injuries, such as ischemia/reperfusion or other acute stresses [26,27]. We should point out that in our experiments, we were testing female and male rats for hypoxia while exercising. This double challenge might have resulted in a focused impairment at complex II in female SSM. On the other hand, it seems that during acute hypoxia, testosterone could have a protective effect on the regulation of blood pressure by acting on the nuclear respiratory factor 1 (NRF-1) a transcription factor [28]. NRF-1 is also involved in both mitochondrial biogenesis and the expression of proteins involved in OXPHOS [28,29]. It could be interesting to research whether NRF-1 is more stimulated by testosterone in males than by estrogen in females, since it is also regulated by estrogen receptor α [30].

Another discrepancy between male and female mitochondria was observed in IFM. When male subjects were trained in hypoxia, the value of ADP/O, obtained with the complex I substrate, increased. However, no such difference was observed in the females when compared to the controls. This could be attributed to either the more efficient activity of F0F1 ATPase or to increased ROS production. This observation led us to investigate the expression of G0/G1 switch gene 2 (G0S2) and ROS-modified protein content.

Protein expression for G0S2 was positive and statistically significantly higher in both female and male hypoxic IFM, but it was absent in SSM.

The G0/G1 switch gene 2 (G0S2) is a protein of about 103 amino acids that was first discovered in mononuclear blood cells, where it was initially believed to unblock cells from the G0 to the G1 phase, although it was later recognized that it does the opposite: maintaining cells in a quiescent state, acting as a tumor suppressor gene in cancer, and regulating adipogenesis [31]. In skeletal muscles, G0S2 is an inhibitor of adipose triglyceride lipase (ATGL), and when it is increased, it induces fatty acid storage in muscle fibers [32]. G0S2 has been associated with mitochondria, and in the heart, it is a hypoxia-inducible factor that improves the efficiency of F0F1 ATPase activity [33]. It seems, that in zebrafish hearts, it induces ischemic tolerance, being a positive regulator of OXPHOS and inducing higher ATP content in mitochondria which overexpress G0S2 [34].

Therefore, the increased protein expression of G0S2 might compensate for the probable impairment of IFM OXPHOS after hypoxia and help to maintain functional mitochondria. It may be that in females, this effect is more reversible than in males, and this would explain why we did not record a change in ADP/O ratio in both sexes with complex I substrates.

The immunoblotting of 4HNE-proteins revealed an increase in ROS in SSM only after acute hypoxic training. The lack of ROS change in IFM confirms what was reported in chronic hypoxia [22]. SSM increase in ROS was statistically significant in males only, and this is in agreement with the better antioxidant capacity of female mitochondria [27]. Increased ROS production has been associated with hypoxia [11,35,36]. Apparently, when oxygen runs short, electrons slip from the electron chain and bind to O_2_, producing superoxide ion O_2_^·−^. It seems that hypoxia induces supercomplex disassembly, thus lower oxygen concentrations are preferably used to produce ROS, instead of ATP [36,37]. In our experiments, acute hypoxia increased ROS in both male and female mitochondria seemingly without causing damage. Interestingly, the yield of SSM mitochondria, which has been linked to mitochondrial content [38], remained unchanged after hypoxia, suggesting a lack of mitophagy.

It is noteworthy that ROS production secondary to hypoxia should elicit hypoxia-inducible factor 1 (HIF-1) release [35], which in turn can lead to deleterious events such as mitophagy and mitochondria fragmentation. Even if we cannot completely rule out the possibility that low concentrations of HIF could be generated after 30 min of training in hypoxia, it seems unlikely that HIF-1 or HIF-2 would be expressed in our samples, as they are typically only expressed after 4–8 h of hypoxia. [39]. Moreover, it has been reported that in the first 20 min of hypoxia, cardiomyocytes produce less ROS than other types of cells, such as endothelial cells, thus suggesting a lesser production of HIF-1 in these cells at shorter times [40].

Following hypoxia, in male brain mitochondria, we observed an enhanced ETS efficiency at complex II and IV. This fact could compensate for any impairment at the single complexes and ensure a regular OXPHOS. Different brain areas can respond differently to hypoxia. Isolated cells from the thalamus after 15 min of hypoxia decrease OXPHOS driven by both complex I and II substrates [14], while the cerebral cortex and hippocampus after 48 h of hypobaric hypoxia have different OXPHOS activities at complex I: the first decreases and the second remains unaltered [10]. In the cerebral cortex, it was also shown that, after hypoxia, complex IV enzymatic activity increased [10]. Since our mitochondria mainly originate from the cerebral cortex, we might assume that at this site in males, ETS increased its efficiency as early as 30 min into hypoxia. All the cited references are related to male rats and no data are available on females. But this difference is worth investigating further.

In our study, it appears that acute hypoxia has a negative impact on the heart, particularly in female mitochondria, while male mitochondria are relatively unaffected. Despite sharing a maternal origin, male and female mitochondria exhibit sex- and gender-related differences in their activities [26]. In fact, about 1000 mitochondrial proteins are encoded by the nuclear genome and, therefore, are subject to hormonal influence in their expression [26]. Research showed that there is not a favored sex, but depending on pathologies, either one might display a higher vulnerability. For instance, females have a higher incidence of heart failure with preserved ejection fraction because, unlike males, they produce less mtDNA and fewer mitochondrial proteins, which are correlated to better diastolic function [41]. On the other hand, female ventricular cardiomyocytes produce less ROS and accumulate less calcium compared to those of males, because they have a higher expression of a supercomplex assembly factor, making female cardiomyocyte mitochondria less susceptible to apoptosis [42]. As we reported above, in acute hypoxia, testosterone may play a significant role in maintaining blood pressure and possibly guarding against mitochondrial damage [28,29]. In this context, our findings are not surprising and can be attributed to the multifaceted activity of mitochondria.

One of the limitations of our study is that we carried out ex vivo studies. In this context, hypoxia was applied for thirty minutes in vivo and then heart and brain mitochondria were isolated and tested in the oxygraph. Thus, our data do not show an impairment concomitant with hypoxia, but, rather, the semi-permanent or, better, prolonged damage of mitochondria induced by short-term exposure to hypoxia. Sometimes, it was hard to find a comparison with other studies, due to the prevalence of in vitro investigations, where hypoxia can be promptly induced and its impact on mitochondria or hypoxia-inducible factors instantaneously scrutinized.

## 5. Conclusions

In summary, our study highlighted some of the effects of acute hypoxia training through an animal model. Recently, analogous acute effects were investigated on human male subjects during training. The findings evidenced defects in oxygen saturation and cerebral oxygenation rather than hemodynamic impairments [7]. We have found that in the heart, only SSM female mitochondria are impaired at complex II. In the brain, male mitochondria hypoxia induced the better efficiency of ETS at complex II and IV. This effect could be comparable to the beneficial effect attributed to chronic hypoxia training [2,6,43]. However, attention must be paid to female subjects exercising under hypoxia, as their heart mitochondria are more vulnerable than those of male subjects.

## Figures and Tables

**Figure 1 biomedicines-11-03149-f001:**
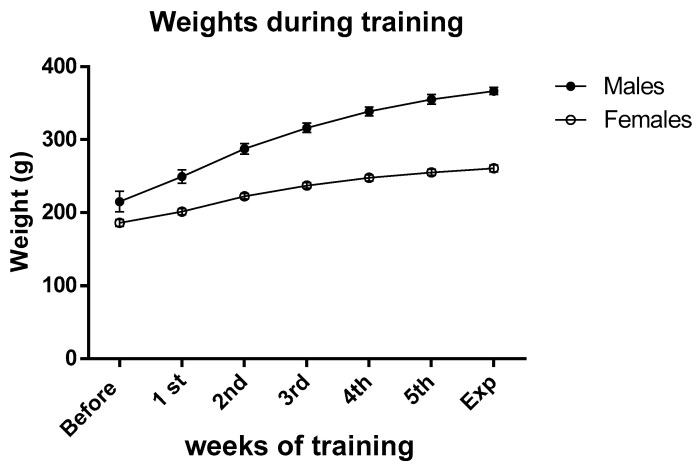
Average weights of male and female rats before and during the training period. Data are the means ± SEM.

**Figure 2 biomedicines-11-03149-f002:**
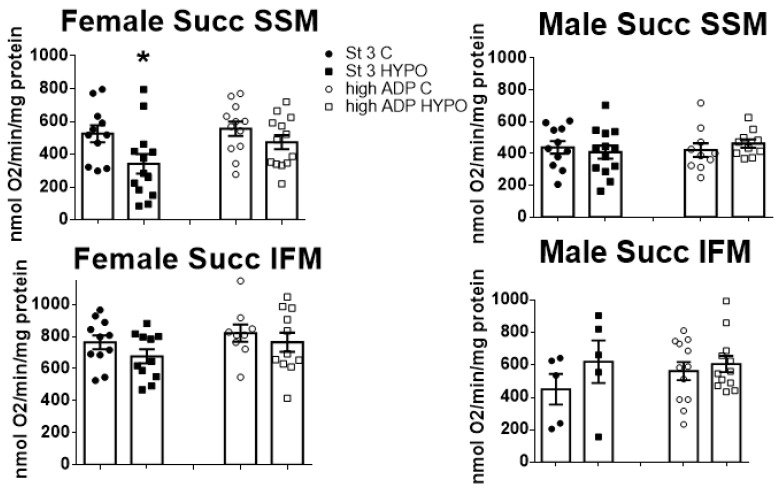
Complex II-related OXPHOS in male and female SSM and IFM. OXPHOS was stimulated via the addition of Succinate (preceded by rotenone administration), followed either by 200 µM ADP (for state 3) or 2 mM ADP (high ADP for maximal oxidative capacity). Note a significant decrease in state 3 of female SSM. * *p* < 0.05, compared to the controls (*t*-test). Data are means ± SEM, *n* = 11–13.

**Figure 3 biomedicines-11-03149-f003:**
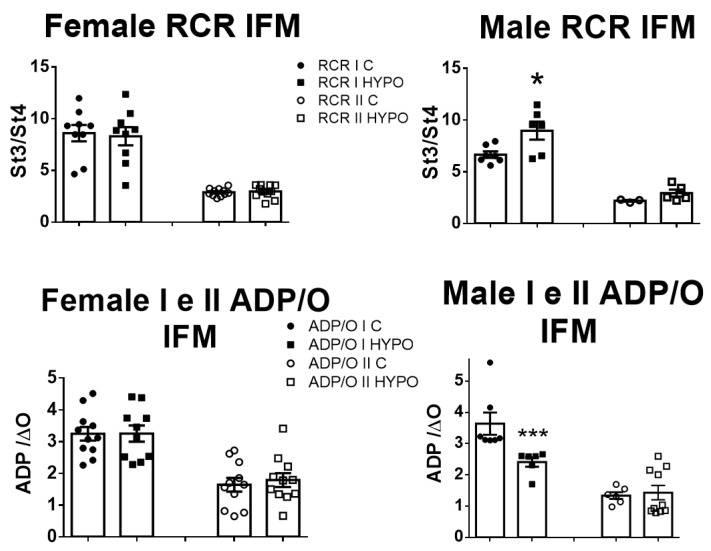
Respiration control ratio (RCR) and ADP/O in female and male IFM with complex I and II substrates. After training in hypoxia, RCR increased in complex I in males only. ADP/O data show a significant decrease in male complex I. * *p* < 0.05, *** *p* < 0.001, compared to control (Mann–Whitney); *n* = 11–12 for complex I and 7–6 for complex II. Data are means ± SEM.

**Figure 4 biomedicines-11-03149-f004:**
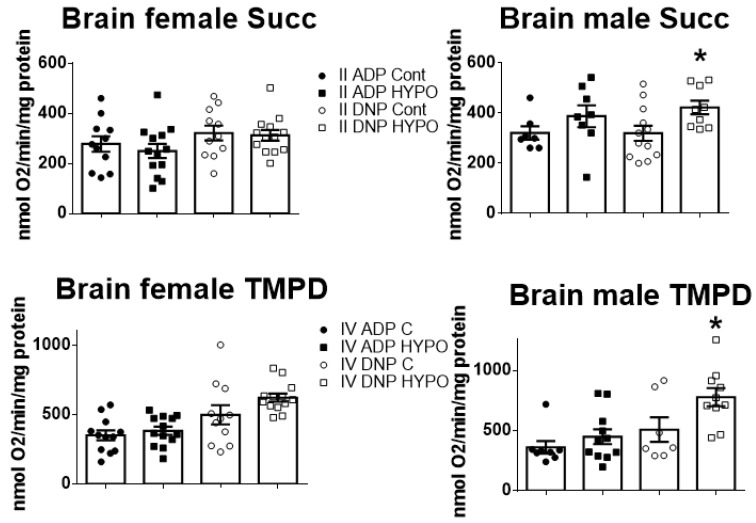
OXPHOS in complex II and IV of isolated brain mitochondria. In complex II, OXPHOS was stimulated by the addition of succinate (preceded by rotenone administration), followed by the addition of 2 mM ADP and 200 µM DNP. For complex IV rotenone, the addition of TMPD, ascorbate, and 2 mM ADP were followed by the addition of 200 µM DNP. Data show that after hypoxic training there is a significant increase in uncoupled OXPHOS at complex II and IV in males only. * *p* < 0.05, compared to controls (*t*-test). Data are means ± SEM; *n* = 8–12.

**Figure 5 biomedicines-11-03149-f005:**
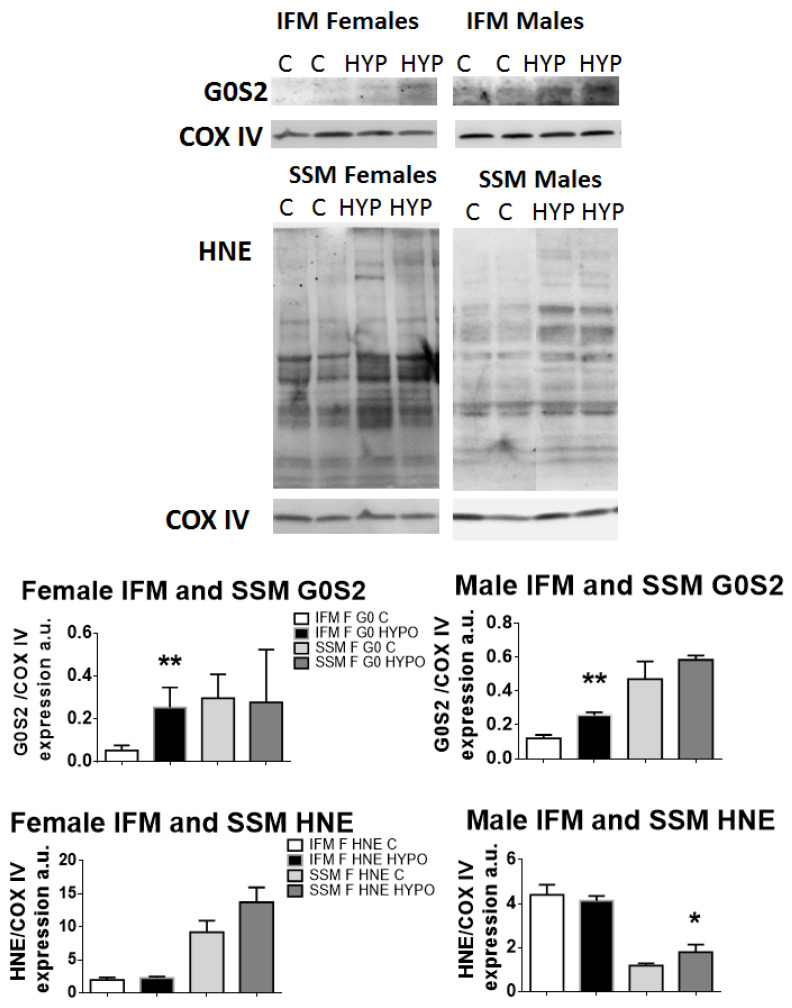
Immunoblotting and band densitometry related to the protein expression of the hypoxia-induced factor G0S2 and of HNE-conjugated proteins. Graphs show an increase in G0S2 in IFM for both sexes and an increase in ROS (HNE) production in SSM for both sexes as well. *t*-test, * *p* < 0.05, ** *p* < 0.01, compared to controls. Data are means ± SEM; *n* = 4–7.

**Table 1 biomedicines-11-03149-t001:** Yields (mg of mitochondrial proteins/g of original tissue) of the different categories of mitochondria in male and female hearts and brains. IFM = heart interfibrillar mitochondria; SSM = heart subsarcolemmal mitochondria. Data represent means ± standard deviation, *n* = 10–17 for brain, and 12–15 for heart mitochondria. Data were evaluated via the *t*-test or the Mann–Whitney U test depending on the Gaussian distribution of the data. None of the data were statistically significant.

Type of Mitochondria	Normoxia	Hypoxia
** *Male* **		
IFM	8.17 ± 2.54	7.45 ± 1.91
SSM	6.73 ± 1.93	7.15 ± 1.30
Brain	8.66 ± 3.14	7.43 ± 2.82
** *Female* **		
IFM	8.85 ± 2.46	7.76 ± 2.01
SSM	7.22 ± 1.91	7.03 ± 2.42
Brain	7.23 ± 1.53	6.82 ± 0.75

## Data Availability

The corresponding author agrees to make available all data generated or analyzed during this study on reasonable request.

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
