# Peer review of "Male and Female Mitochondria Respond Differently after Exercising in Acute Hypoxia"

_biomedicines, 2023, doi:10.3390/biomedicines11123149_

Round 1

Reviewer 1 Report

Comments and Suggestions for Authors

SUMMARY:

Authors report on the effects of acute (30 min) hypoxic exposure under normobaric pressure, under conditions of high aerobic demand, on mitochondria isolated from rat heat and brain tissue. Rats had previously undergone a 5-week normoxic, normobaric training regimen. Authors find small differences in a few tissues that, although statistically significant, are not convincingly meaningful. The inaccurate descriptions of basic bioenergetic measurements (ADP/O, RCR) raise concerns about data presentation and interpretation. An interesting idea is tested, but a clear rationale for what changes to mitochondrial activity may be expected by the very short intervention, followed by a return to normoxia, are not clear. There are enough questions about the research design and interpretation to make the findings uninterpretable. For these reasons, this manuscript is not publishable at this time.

MAJOR COMMENTS:

Fig 2: the significant difference noted in Fig 2 may be statistically detectable according to the reported values, but does not seem convincing given the apparently noisier (higher variation) measurements- a power analysis showing the likelihood of the number of animals used demonstrating a meaningfully reproducible difference would be necessary to help to contextualize this finding- is the population size used sufficient to make the claim that succinate-driven respiration by SSM mitochondria is reproducibly or meaningfully different?

Line 215, Figure 3: the ADP/O is a molar ratio (mol ADP phosphorylated/mol O consumed), not a mol/rate ratio as described in the text.

Line 222: states that increased RCR indicates “increased oxygen consumption rate when stimulated” – this is not what RCR indicates. Higher RCR (phosphorylating OCR + leak OCR)/(leak OCR) indicates better coupling, which is better described by the coupling efficiency (state 3-state 4)/mitochondrial OCR. Morever, RCR is typically used to understand the quality of the mitochondrial preparation, and may or may not reflect the mitochondrial status prior to isolation (see, e.g., Brand and Nicholls 2011 doi: 10.1042/BJ20110162.)

Figure 4: The representative blot images shown raise concerns that the background signal is contributing to the reported intensities across the 4-7 blots analyzed. How background subtraction is handled is not described in the manuscript.

MINOR COMMENTS:

Intro:

 Line 68: “FiO2” needs to be defined and used consistently.

Line 185: assertions not substantiated with data

Line 187: Fig 1 shows an increase in body weight for which no statistical analysis is shown; text says “within expected range” but this is not inconsistent with a statistically significant increase in body weight over the training period- this should be evaluated

Line 198: though methods describe general approach, table 1 does not specify which statistical test was used

Figure captions do not always explain the labels used in the figures (e.g., Fig 2, “ADP100”)

In Figure 3, are the ADP/O values for both Complex I and Complex II substrates combined?

Methods:

Please give age of rats used in trials (it appears in results but should be included with animal description in Methods).

The number of individual data points in different measurements varies; clearer description of how many animals were in each treatment group and inclusion/exclusion criteria for the final measurements reported is necessary to understand why.

Comments on the Quality of English Language

English syntax and grammar errors make the meaning of many parts of the manuscript difficult to understand and creates uncertainty in how many statements should be interpreted. 

Reviewer 2 Report

Comments and Suggestions for Authors

1.I think the study design and conclusion of this manuscript are logical

2.I only have a sugession of adding sex hormone effects on mitochondria in the part of discussion

Round 2

Reviewer 1 Report

Comments and Suggestions for Authors

The manuscript has improved; though some concerns remain:

MAJOR:

The major concern is that the study in underpowered, weakening the conclusions made. Authors state in their response letter that “If I perform an a priori analysis with GPower software and put effect size 0.94 (as it comes from calculations of my “n” number, average and SD), α 0.05, and power 0.8, the ideal size of “n” should be 19.Yet the n for the isolated mitochondrial experiments is 8-12. I disagree with the authors’ statement thata power of a test is of considerable interest in order to assess the probability of false negatives” but, a posteriori, it does not concern the reliability of significant, positive results.” – proper power analysis assesses the likelihood of both Type 1 (false positive) and Type 2 (false negative) errors.

MINOR:

In response to the ADP/O data in Figure 3:  Given that ADP/O was calculated correctly despite the original description (but corrected in the revision), the values shown in Fig 3 raise questions. The P/O values greater than 3 are unlikely given oxidation of pyruvate and malate, while the value between 2 and 3 (shown for the ADP/O I HYPO male sample mean) is more plausible. See e.g., Hinkle and Yu, 1979: THE JOURNAL OF BIOLIGICAL CHEMISTRY 254:2450-2455. This could be e.g., a technical offset due to an instrument calibration issue, but should be addressed.

Clarity of writing throughout: has been minimally addressed. 

Comments on the Quality of English Language

Clarity of writing throughout: has been minimally addressed. 
